# Embedded Ultrasonic Inspection on the Mechanical Properties of Cold Region Ice under Varying Temperatures

**DOI:** 10.3390/s23136045

**Published:** 2023-06-29

**Authors:** Huimin Han, Li Wei, Nizar Faisal Alkayem, Maosen Cao

**Affiliations:** 1College of Mechanics and Materials, Hohai University, Nanjing 210024, China; hanhm9011@hhu.edu.cn (H.H.); weili0019@hhu.edu.cn (L.W.); 2College of Water Conservancy and Hydropower Engineering, Hohai University, Nanjing 210098, China; nizar-alkayem@hhu.edu.cn; 3College of Civil and Transportation Engineering, Hohai University, Nanjing 210098, China

**Keywords:** ice, mechanical properties, embedded ultrasonic system, varying temperature, random pore model, porosity

## Abstract

The mechanical properties of ice in cold regions are significantly affected by the variation in temperature. The existing methods to determine ice properties commonly rely on one-off and destructive compression and strength experiments, which are unable to acquire the varying properties of ice due to temperature variations. To this end, an embedded ultrasonic system is proposed to inspect the mechanical properties of ice in an online and real-time mode. With this system, ultrasonic experiments are conducted to testify to the validity of the system in continuously inspecting the mechanical properties of ice and, in particular, to verify its capabilities to obtain ice properties for various temperature conditions. As an extension of the experiment, an associated refined numerical model is elaborated by mimicking the number, size, and agglomeration of bubbles using a stochastic distribution. This system can continuously record the wave propagation velocity in the ice, giving rise to ice properties through the intrinsic mechanics relationship. In addition, this model facilitates having insights into the effect of properties, e.g., porosity, on ice properties. The proposed embedded ultrasonic system largely outperforms the existing methods to obtain ice properties, holding promise for developing online and real-time monitoring techniques to assess the ice condition closely related to structures in cold regions.

## 1. Introduction

In recent decades, research on ice has grown remarkably. Intensive research is being performed on whether to examine global climate change [1,2,3], to break ice sheets in the arctic with icebreakers [4], to investigate ice-covered electric antennas [5], to design ships and offshore structures [6,7], to study the icing conditions in aviation [8], etc. In those investigations, it is essential to obtain the mechanical properties of ice, which will change with variations in ambient temperature.

Currently, diverse measurement methods of ice mechanical properties have been developed in an experimental way, including uniaxial compression [9,10,11], triaxial compression, and flexural strength tests [12]. Aksenov et al. [13] attained the preliminary temperature-related stress-strain properties of the freshwater ice by performing short-time uniaxial compression tests for cylindrical ice specimens at various specific temperatures. Moslet [14] conducted the uniaxial compression strength tests on columnar sea ice in the field on Svalbard, with the indication of a strong relationship between Young’s modulus and ice porosity. Qiu et al. [15] obtained the compressive and tensile plastic properties of ice based on the triaxial compression test of columnar ice. However, due to the brittleness of the ice material itself, fracture is inevitable in the experiment. Moreover, the time between the testing starting and the fracture ending is very short. Therefore, in a single experiment, it is not possible to obtain changes in ice properties caused by continuous temperature changes. Additionally, those destructive tests are costly and inappropriate for structures in service. In order to obtain the mechanical properties of ice that continuously change with temperature without damaging the existing ice structure, a non-destructive testing method is needed that can achieve real-time monitoring and is suitable for ice materials. Accordingly, a non-destructive method for estimating the temperature-related mechanical properties of ice is demanded.

The ultrasonic technique is one of the most commonly used non-destructive methods in characterizations of material properties [16,17], derivations of dynamic responses [18], and condition monitoring [19,20,21,22,23]. Relevant sensing techniques for wave generation and measurement have been developed rapidly. Representative research works are as follows: Bayón et al. [24] employed the measured Rayleigh wave velocity and the aspect ratio of the elliptical trajectory amplitudes to obtain the Young’s constants of isotropic linear materials. Further, Medina and Bayón [25] proposed a method for calculating the dynamic Young’s constants of an anisotropic plate through the measured impact-echo resonance and Rayleigh wave velocity. Medina and Bayón [25] determined the mechanical properties and damage properties of a multilayer composite board by combining experimental and numerical data simultaneously. Ultrasonic technology has also been implemented in the estimation of ice properties. Bock and Polach [26] explored the applicability of the collected long-period surface wave dispersions in the inversion of ice shell thickness. Gagnon [27] presented an impulse-echo method to evaluate the longitudinal ultrasonic velocities of three different types of specimens to calculate Young’s modulus. Noteworthy, a state-of-the-art literature review specified that only a limited number of studies dealt with the utilisation of ultrasonic techniques in the determination of ice properties, and there are still some crucial issues to be solved. In addition, after obtaining the monitoring data, this study used the methods mentioned in [28,29] to conduct the corresponding data modelling and analysis, including regression analysis and spectral analysis.

In addition to experimental testing methods, numerical simulation is a valuable alternative method for estimating the ice properties by taking advantage of wave propagation simulation techniques [22,30,31,32,33]. Various numerical models were established to represent the ice materials, including Young’s non-linear behaviour model [34], the crushable foam model [35,36], and the user-defined elastic-plastic material model [37,38]. Liu et al. [6] proposed a quasi-static model on the basis of the strain rate-dependent plasticity theory to simulate ice structural behaviour by combining the Tsai Wu yield surface criteria and the associated flow rule. Bock and Polach [26] analysed the non-linear behaviour of the aqueous model ice, concluding that the non-linear behaviour of ice is independent of its crystal structure and chemical dopant. Gagnon et al. [27] adopted a “crushable foam” material type in LS-DYNA to model the ice behaviour, with the numerical results of the ‘calibrated stress-volumetric strain relationship’ showing a good agreement with the experimental ones. Noticeably, the existing numerical cases lacked studies on the influence of temperature variations and ice porosities.

From the surveyed literature, it is observed that the current prevalent methods to determine ice properties commonly rely on one-off and destructive compression and strength tests, which are incapable of acquiring the temperature variation-induced changes in ice properties. Moreover, ultrasonic technology, as one of the most commonly used non-destructive methods for characterising material properties, is rarely applied to ice mechanical properties. In addition, due to the unique nature of ice materials, it is necessary to study how ultrasonic sensors can be embedded in ice. Furthermore, there is a lack of numerical models for wave propagation in ice with different temperatures and porosities. To address these issues, an embedded ultrasound system is proposed to test the mechanical properties of ice, and its feasibility and effectiveness have been verified through ultrasound experiments and numerical models.

The rest of this paper is organised as follows: Section 2 formulates a novel ultrasonic method for determining the mechanical properties of ice. Section 3 builds the 2D and 3D ice models, with the numerical results at different temperatures and porosities being counted. Section 4 exhibits comparisons and discussions of experimental and numerical results. Section 5 presents the conclusions.

## 2. Test-Based Identification of the Mechanical Properties of Ice

The current prevalent methods to acquire the mechanical properties of ice are typically uniaxial compression, triaxial compression, and flexural strength experiments. However, due to the brittleness of the ice material itself, fracture is inevitable in the experiment. Moreover, the time between the testing starting and the fracture ending is very short. Therefore, in a single experiment, it is not possible to obtain changes in ice properties caused by continuous temperature changes. To address the deficiencies of existing methods, an embedded ultrasonic system is built as an alternative to compression tests to obtain the mechanical properties of ice at different temperatures.

### 2.1. Test Specimen

#### 2.1.1. Actuators and Sensors

Piezoelectric material can be used as both actuators and transducers, which is beneficial for its direct and inverse piezoelectric effects [39]. Figure 1c shows a piezoelectric ceramic transducer (PZT), with its dimensions and material properties shown in Table 1. In the PZT, the electric wires are welded to the positive and negative poles on their different sides. Both sides are waterproof with the transparent insulating glues. The temperature sensor is made of a 4 mm-diameter, waterproof platinum probe (shown in Figure 1d). The measurement range is −50–+200 °C.

#### 2.1.2. Ice Specimen and Sensing Strategy

(1)Ice specimen

The ice specimen is fabricated using distilled water with the purpose of avoiding excessive impurities. The dimensions of the ice specimen are 150 × 150 × 200 mm^3^. Before freezing, all the required sensors during the ultrasonic tests are fixed in the middle of the mould with steel wires, as shown in Figure 1b.

The process for making ice samples is as follows: first, place the mould with distilled water and sensors into a refrigerator with an initial temperature of −5 °C; second, when the surface water is frozen, progressively decrease the temperature of the refrigerator by 5 °C per two hours; next, when the refrigerator temperature reaches −35 °C, the ice specimen should be fully made; and finally, the newly-made ice specimen needs to be stored in a refrigerator at −35 °C for at least 48 h to ensure a uniform temperature distribution in the ice specimen. In this way, the ice cracks caused by big temperature variations could be avoided.

(2)Sensing strategy

The PZTs and temperature sensor are set in the middle of the ice specimen. The distance between the two PZTs is considered to be 70 mm. Additionally, the temperature sensor is set at the uniform level of the PZTs due to the effect of temperature stratification in ice.

(3)Considerations

Preliminary tests with a distance between the two PZTs are conducted to determine the appropriate properties. Comparing the results at 25 mm, 50 mm, and 70 mm, it is found that when the distance between the two PZTs is small, the wave packets of the longitudinal wave and the shear wave cannot be distinguished. The reason is that the propagation speed of the wave is very fast. When the shear wave propagates, the longitudinal wave has not been fully received, which causes the longitudinal wave and shear wave to be superimposed on each other, so it is difficult to distinguish a separate shear wave packet.

### 2.2. Velocity of the Transmitted Waves

Wave velocity, one of the key properties of wave propagation research, is defined as the velocity at which a disturbance propagates in specified materials. It mainly depends on the material properties, structural geometries, and external excitations. The longitudinal and shear wave velocities are the most widely used variables in ultrasonic-structural analysis. Their relationships with structural properties are formulated as follows:(1)$$V_{l}=\sqrt{\frac{E\left(1-\mu \right)}{\rho \left(1+\mu \right)\left(1-2\mu \right)}},$$
(2)$$V_{s}=\sqrt{\frac{E}{2\rho \left(1+\mu \right)}}$$

According to Equations (1) and (2), the material properties (such as *E*, *G*, and *μ*) can be achieved with the measured longitudinal and shear wave velocities (*V_l_* and *V_s_*) (shown in Equations (3)–(6)). These relationships, which are the fundamentals of the characterizations of the ice properties, can be written as follows:(3)$$\mu =\frac{V_{l}^{2}-2V_{s}^{2}}{2\left(V_{l}^{2}-V_{s}^{2}\right)},$$
(4)$$E=\frac{V_{s}^{2}\rho \left(3V_{l}^{2}-4V_{s}^{2}\right)}{V_{l}^{2}-V_{s}^{2}},$$
(5)$$G=\rho V_{s}^{2},$$
(6)$$K=\rho \left(V_{l}^{2}-\frac{4}{3}V_{s}^{2}\right),$$
where *ρ* is the density, *E* is Young’s modulus, *μ* is Poisson’s ratio, and *V_l_* and *V_s_* represent the velocity of the longitudinal wave and the shear wave, respectively.

### 2.3. Experimental Setup and Methods

(1)Test process

The experimental apparatus for the ultrasonic test is presented in Figure 2.

The excitation part includes the wave generator and desktop. Firstly, the waveform of excitation is generated on the desktop. Then, a tone burst signal, consisting of sinusoids modulated by the Hanning window [40], is utilised as the external excitation. This signal can be expressed as follows [41]:(7)$$x\left(t\right)=\frac{A}{2}\sin\left(2\pi f_{c}t\right)\left[1-\cos\left(\frac{2\pi f_{c}t}{n}\right)\right],$$
where *A* is the amplitude, *n* denotes the number of signal cycles, and *f_c_* represents the central frequency.

The designated excitation is then transferred into the Agilent 33250A arbitrary wave generator, which converts the excitation from a digital signal to an analogue one. The output terminal of the waveform generator is connected to the PZT through a Bayonet Nut Connector cable. Because of the positive piezoelectric effect, the PZT deforms in the longitudinal direction of the ice, leading to the generation of longitudinal waves. The output voltage of the waveform generator, which regulates the excitation amplitude, is set to be 10 V in order to ensure a fully deformed piezoelectric ceramic sheet.

In the receiving part, the PZT transducer generates current due to the inverse piezoelectric effect. Based on the characteristics of PZT, this generated current has a linear relationship with its deformation. Thus, the captured current from the receiver can be regarded as the propagated longitudinal wave in the ice. This received current is then amplified by a fixed-gain universal preamplifier, PXPA3, to increase its amplitude. The transfer gain of the charge amplifier is 10 mv/pc. Then the current is introduced into the Agilent DSO7034B oscilloscope to be converted into a digital signal. The sampling frequency of the oscilloscope is set to 50 MHz. In particular, the application of the waterproof glue on the surface of PZT not only makes the sensor insulated but also prevents the generated current from leaking into the ice. In addition, there is no extra circuit other than the experimental equipment. Therefore, the measured current from the receiver thoroughly originates from the inverse piezoelectric effect caused by the longitudinal wave.

During the ultrasonic test, the ice specimen is covered with a foam box, which is a common thermal insulation material in daily life and is used to slow down the impact of external temperature on water, thus slowing down the speed of water icing. If the freezing speed is too fast, it can cause the ice surface to expand and crack. Cracking ice will badly affect the experimental results. The internal temperature is monitored in real-time through the temperature sensor. By continuously recording these internal temperatures and the corresponding received signals, the non-destructive monitoring of wave propagation in ice under the condition of temperature variation is realised.

(2)Determination of signal properties

Two properties (*n* and *f*_c_) on the right-hand side of Equation (7) need to be determined in advance. Preliminary tests with different *n* and *f*_c_ are conducted to determine the appropriate properties. The measured ultrasonic signals with different *n* and *f*_c_ are shown in Figure 3 and Figure 4, respectively.

In the preliminary test, the response signal results of 2 periods, 3 periods, 4 periods, and 5 periods are compared and analysed. The experimental results in Figure 3 indicate that the second peak of the measured signals can be more easily distinguished in cases of having a smaller number of cycles (*n* = 2, 3) than in cases of having a larger number of cycles (*n* = 4, 5).

On the other hand, the results in Figure 4 imply that the second peak of the measured signals is easier to identify in cases of having larger central frequencies (*f_c_* = 250 kHz, 300 kHz, 350 kHz, 400 kHz, 450 kHz, and 500 kHz) than in cases of having smaller central frequencies (*f_c_* = 150 kHz and 200 kHz). 

Accordingly, a normalised 2-cycle 250 kHz tone burst signal (shown in Figure 5) is employed as the external excitation in the experimental tests below.

### 2.4. Experimental Results

#### 2.4.1. The Threshold Denoising Based on the Wavelet Transform

The mode reflection/conversion is inevitable at the external and internal boundaries of the ice specimen, contributing to a complex multi-mode wave signal. Moreover, the measured signals could be adversely affected by environmental noise. Therefore, it is necessary to rectify the measured signals to extract the inherent characteristics of the interested signal rather than those induced by mode reflection/conversion and noise. In this study, a wavelet transform-based threshold denoising process is adopted, with the results presented in Figure 6.

#### 2.4.2. Experimental Results

Considering the faster propagation of the longitudinal wave compared with the shear wave, the first and second peaks of the ultrasonic waveform can be regarded as the longitudinal wave and the shear wave, respectively. Figure 7 shows four denoised ultrasonic waveforms measured at different temperatures. According to Figure 7, the first peak of the presented waveforms cannot be clearly identified. Under these circumstances, the first wave trough of the excitation cycle is selected as a reference to calculate the propagation velocities of longitudinal and shear waves in the ice. Correspondingly, the selected troughs of the longitudinal and transverse waves are annotated by the green circleabaquss in Figure 7.

Figure 8 presents the calculated wave velocities with temperatures ranging from −35 °C to −0.5 °C. Evidently, the velocities of longitudinal and shear waves are strongly related to temperature. To explicitly interpret the relationship between temperature and wave velocities, two quadratic functions are utilised to fit the velocity-temperature curves in Figure 8, respectively, which are:(8)$$V_{l}=-0.01139\;T^{2}-4.36647\;T+3783.56,\;R=0.99598,$$
(9)$$V_{s}=-0.00401\;T^{2}-1.14069\;T+1797.705,\;R=0.99449,$$
where T represents the temperature and R is the correlation coefficient.

Table 2 shows the longitudinal wave velocity in ice obtained from existing research. The velocity values are in good agreement, which validates the accuracy of experimental results in this study. The wave velocity and Young’s modulus strongly depend on properties (such as salinity, impurity content, etc.) during the production of the ice, and the influences of temperature and time are not taken into account in these experiments. This may be the reason behind the deviation from the values, such as the distance of the transducers measured in the present experiment.

In the experiments, the density of the utilised ice specimen was 890 kg/m^3^, which is the average value determined by the drainage method [45,46,47]. Combined with Equations (3)–(6), the mechanical properties of ice at different temperatures can be achieved, including the Young’s modulus, shear modulus, Poisson’s ratio, and bulk modulus. The recognition results are exhibited in Figure 9, with the vertical axis representing the corresponding properties and the horizontal axis signifying the temperature variations. All the properties presented in Figure 9 decline continuously with increasing temperature.

## 3. Physics-Directed Numerical Simulations

In nature, ice contains pores, which are related to the growth process of ice. The formation of ice from water is a process from the outside to the inside. After the surface freezes, the air in the water cannot pass through the ice, forming pores inside the ice. These pores vary in size and are randomly located, so randomly distributed circular pores are used in the manuscript to simulate the pores inside the ice. The dimension, distribution, and amount of the entrapped air bubbles highly affect the dynamic mechanical properties of ice. Hou et al. [48] indicted that those bubbles in ice could neither be manufactured artificially nor obtained in the desired amount and distribution. Therefore, the influence of the pores in the ice on the mechanical properties can only be studied by numerical methods. In this study, a stochastic algorithm is presented to produce the sparsely distributed air bubbles in the numerical model of ice.

### 3.1. Material Properties of the Ice Model

In order to better compare the numerical and experimental results, four sets of experimental results were selected as the model material’s mechanical properties. The specific values are shown in Table 3.

### 3.2. Two-Dimensional Ice Random Pore Model

In the numerical model, ice is treated as a linear solid without considering its nonlinear behaviour or tensile or compressive damage. The dimensions of the tested specimens are 210 × 210 mm^2^, and the pores (air bubbles) are concentrated within a central region of 180 × 180 mm^2^. Additionally, the round pores in the 2D model are utilised. Before establishing the model, we observed and measured a large number of pores in natural and artificial ice. These pores, except for those larger than 5 mm on the surface of the ice, are very small in size inside the ice, some of which cannot be measured using conventional measurement methods. In order to restore the uniformly distributed pores accumulated in the ice, the porosity during simulation was fixed, the distribution was selected from the range of most pore sizes, and the appropriate pore distribution was calculated using MATLAB R2018a. They are classified into small pores having diameters of 0.5–1 mm and large pores having diameters of 1–1.5 mm, with the volume ratios being 6:4. This volume ratio was obtained through extensive testing, which not only meets the requirements of achieving porosity within the model but also achieves a uniform distribution of pores within the model. Moreover, the pores in the ice model are reciprocally independent, with distances between the pores greater than 1.5 mm. In the modelling process, random pores are first generated using MATLAB R2018a, and then the generated pore model is imported into the ice model built in ABAQUS 6.14. The final model is obtained by cutting out random pores from the original ice model. In the model, the excitation of the PZT actuator is simplified by applying a concentrated force at the same position as the PZT sensor [49]. The concentrated force is generated by the same 2-cycle 250 kHz tone burst signal as the experimental test (as shown in Figure 5). In addition, the distance between the actuator and receiver is 70 mm. In the model, the boundaries around the ice are the boundaries of wave propagation.

The porosity is obtained using the calculation method of Cox and Weeks [50]. The specific method can be expressed as:(10)$$v_{a}=1-\frac{\rho }{\rho_{i}\left(T_{i}\right)}+\rho S_{i}\frac{F_{2}\left(T_{i}\right)}{F_{1}\left(T_{i}\right)},$$

The density of pure ice $\rho_{i}$ (g/cm^3^) is described as a function of temperature as [49]:(11)$$\rho_{i}\left(T_{i}\right)=0.917-1.403\times 10^{-4}T_{i},$$
where $\;v_{a}\;$ is the air volume ratio, $\rho_{i}$ is the pure ice density, $S_{i}$ is ice salinity, and $T_{i}$ is the ice temperature. In this study, distilled water is used to make ice samples, so in the numerical model $S_{i\;}$ = 0 is adopted. Therefore, Equation (10) becomes:(12)$$v_{a}=1-\frac{\rho }{\rho_{i}\left(T_{i}\right)}.$$

In the established numerical model, the density of the ice sample obtained from the experiment is employed to calculate the porosity at different temperatures following Equations (10)–(12).

Three 2D ice models (shown in Figure 10 and Figure 11) with different porosities (0.5%, 1%, and 3%) were established to investigate the influence of porosities on the wave propagations in ice. The white dots in Figure 10 and Figure 11 represent pores within the ice, while the green areas in Figure 10 represent the ice. The blue part in Figure 11 shows the grid inside ice. The mesh size is 1 mm, and the time step of numerical integration is 2 × 10^−8^ s. These small computational properties can ensure the accurate capture of the propagative behaviour of the ultrasonic wave.

### 3.3. Numerical Simulations of 2D Ice Random Pore Model

#### 3.3.1. Wave Propagation Analysis with 0.5% Porosity

Figure 12 shows the wave propagation process of the ice model with 0.5% porosity at different time values. Both the longitudinal and shear waves can be clearly distinguished within a short period of time after being excited. The red circle in Figure 12d shows that waves are superimposed on each other during propagation, and various waves cannot be separated. Figure 12e shows a horizontal rebound wave at the red circle. The whole specimen in Figure 12f is filled with mixed waveforms, and it is impossible to distinguish between shear waves and longitudinal waves.

The waveforms received by the response point at four temperatures are extracted, and the drawn waveform is shown in Figure 13. As can be seen in Figure 13, there is no significant change in the morphology of the waves at the four temperatures. If the image is not enlarged, it is difficult to detect the difference between the four curves. Therefore, we believe that the change in temperature has little effect on the waveform. However, in the red dashed circle in Figure 13, the results of amplifying the peaks show that the time of the peaks appearing in the four curves is different, which means that the wave propagation speeds corresponding to the four temperatures are different (because the distance between the sensors remains constant).

With the application of ice properties from experiment tests, the numerical results of longitudinal and shear wave velocities at different temperatures can be obtained, as shown in Figure 14.

#### 3.3.2. Non-Porous 2D Ice Model

To reveal the existence of pores on the wave propagation in ice, this section compares the numerical results attained by a non-porous ice model and a model with 3% porosity. The result of −17.2 °C is randomly selected to demonstrate the process of wave propagation in ice in numerical simulations. Of course, other temperature results can also be selected for display. The selection of different temperatures has no effect on the propagation process of waves in ice. The comparison results of wave propagations are depicted in Figure 15 and Figure 16. In Figure 15b,c, the longitudinal and shear waves can be apparently inspected. According to the non-porous ice models in Figure 15b–f, no wave superpositions are observed near the excitation point, demonstrating the boundary of the critical rebound effect of the pore boundaries on the wave propagations in ice, even when the pore size is small.

Figure 17 presents the normalized waveforms of the response point. The circles represent the positions of the peaks used in velocity calculation, with the red circles indicating the peaks used for calculating the longitudinal wave velocity and the green circles used for calculating the shear wave velocity. According to Figure 17, the presence or absence of pores obviously affects the wave propagation speed. In Figure 17, the amplitude in the first wave packet of the non-porous ice model is greater than that of the porous ice model. However, in the range of 0.4–0.5 s, the amplitude of the porous ice model is greater than that of the non-porous ice model. This is because the pore size in the porous model is 0.5–1 mm, which is much smaller than the wavelength (about 1500 m). Therefore, waves will form multiple reflections, refractions, and diffraction superpositions under the rebound effect of the pore wall.

Figure 18 shows the normalised waveforms of the three porosities at −17.2 °C. The circles in Figure 18 indicate the values used for calculating the wave velocities, with the red circle is used for calculating the shear wave velocity and the green circles are used for calculating the longitudinal wave velocity. It can be seen from Figure 18 that the higher the porosity, the slower the wave propagation velocity.

### 3.4. Numerical Simulation Results of the 3D Ice Random Pore Model

In this section, the 3D random pore ice model is built by expanding the 2D model. The dimensions of the investigated 3D model are 150 × 150 × 50 mm^3^, and the pores are concentrated within a central cuboid range of 140 × 140 × 45 mm^3^. Similar to the 2D model, the random pores in the 3D model are spheres, which are classified into the small size with diameters of 0.5–1 mm and the large size with diameters of 1–1.5 mm. The ratio of small and large pores is 6:4. Each pair of pores has a distance larger than 3 mm. Additionally, the porosity of the 3D model can be designated in accordance with the porosity of the 2D model (shown in Equations (10)–(12)). Three 3D ice models with porosities of 0.5%, 1%, and 3% are counted below.

The coordinates of the centre and the diameter of the pores are randomly generated. When the diameter rotates 360 degrees around the centre, a spherical pore is developed. All pores of the entire model are shown in Figure 19a. By subtracting the pores from the whole ice, the final 3D ice model with random pores, as shown in Figure 19b, can be obtained. In addition, the three columns in Figure 19 represent three model diagrams with different porosities.

The excitation signal in the 3D model adopts the waveform signal of Figure 5 and acts with concentrated force on the excitation point. The distance between the excitation and the response points is 70 mm. The centre frequency of the sine wave is $f_{c}=250$ kHz. As the mesh size of the model is 1 mm, the time step is 2 × 10^−8^ s. The grid map is shown in Figure 20, where Figure 20b is a planned view of the model and Figure 20c is an enlarged picture of the red circle of Figure 20b, which is used to highlight the grid at the gap. The red circles in Figure 20c highlight the pores in the model.

### 3.5. Numerical Results from the 3D Ice Model

#### 3.5.1. Wave Propagation in Ice with 1% Porosity

Figure 21 presents the propagation process of waves in the 3D model. According to Figure 21a–c, within a short period of time after excitation, the longitudinal and shear waves can be clearly distinguished. Nevertheless, as time goes on, the longitudinal and shear waves become bounced and superimposed on each other. Compared with the 2D model, these bounces and superpositions of ultrasonic waves in the 3D model are more serious and complex.

The mechanical properties of ice at various temperatures obtained from the experiment are input into the 3D numerical model as the material properties, and the shear wave together with the longitudinal wave velocities at different temperatures can be obtained. Finally, all wave velocities are plotted as a scatter diagram, as shown in Figure 22. 

Four groups of data are selected from all these numerical outcomes. The waveforms of wave propagation in ice at these four temperatures are shown in Figure 23. It can be seen that the shape of the waveforms at each temperature is similar, and there are obvious differences in the speed of wave propagation at different temperatures. The positions of the red circles in Figure 23 correspond to the first troughs of the longitudinal wave and the shear wave, respectively.

#### 3.5.2. Comparisons

This part investigates the influence of porosity on wave propagation in ice using a 3D model. The normalised waveforms at different porosities are shown in Figure 24. Similar to the results in 2D, the porosity only affects the speed of wave propagation, and the influence on the waveform can be ignored. The circles represent the peak positions used for velocity calculation. Among them, the red circle and the green circle represent the peak values used for calculating longitudinal wave velocity and shear wave velocity, respectively. The cluttered waveforms at the tail (the wave propagation time $\;t\geq 5.5\times 10^{-5}$) in Figure 24 indicate the complex superposition and rebound of waves in the 3D model.

## 4. Discussion of Experimental and Numerical Results

This section compares the numerical and experimental results to explicitly verify the reliability of the established 2D and 3D numerical ice models. Taking the case of temperature −17.2 °C as an example, the corresponding numerical and experimental results are presented in Figure 25. The figure shows that the waveform is normalised. The circles represent the peak positions used for velocity calculation. Among them, the red circle and the green circle represent the peak values used for calculating longitudinal wave velocity and shear wave velocity, respectively. According to Figure 25, both experimental and numerical results can distinguish the transverse wave from the longitudinal wave. Moreover, the waveforms are similar except for a certain difference in magnitude. Table 4 presents the simulation errors of the established 2D and 3D ice models at various temperatures. The relative errors are within the range of 0.15–5.7% according to the comparison of numerical and experimental results. The errors in the 3D model are less than those in the 2D model, which indicates that the 3D numerical ice model is more appropriate for wave propagation analysis of ice.

Table 4 shows that the small deviation range demonstrates the reliability of both the experimental tests and the numerical simulations. Moreover, the mechanical properties of ice obtained from the experimental results are reasonable and applicable. In addition, the 2D and 3D numerical ice models can prove that this method of simulating ice with a random pore model is feasible. The ice model provides a reliable and feasible way for further investigations of ice-related engineering problems.

## 5. Conclusions

In this study, a sophisticated embedded ultrasonic system is proposed to inspect the mechanical properties of ice in real time and online. This embedded ultrasonic system provides a platform to continuously obtain the response of the ice. With this system, the ice properties under specific temperature conditions are identified based on the intrinsic relationships between the wave propagation velocities and the mechanical properties of ice. Furthermore, the feasibility and effectiveness of the proposed embedded ultrasonic system-based method are validated via ultrasonic experiments and numerical simulations. Both numerical and experimental results demonstrate the effectiveness of the proposed embedded ultrasonic system-based method to inspect the mechanical properties of ice. The conclusions can be summarised as follows:
(1)With the artificial ice specimen made of distilled water, the experiment results indicate the velocities of wave propagation in the ice decrease gradually with the elevated temperatures. Moreover, the velocity of the longitudinal wave is within the range of 3954.8–3787.88 m/s, while the velocity of the shear wave is within the range of 1831.54–1797.64 m/s;(2)The experimentally obtained wave velocities have been brought into known formulas to calculate the material properties of ice at different temperatures, including Young’s module, Poisson’s ratio, shear modulus, and bulk modulus. The results have shown that the values of the four properties have a downward trend with the increase in temperature;(3)With the help of the numerical models, the wave propagation in ice has been investigated, and the influence of varying temperatures on the wave propagation has been discussed in detail. The relative errors are within the range of 0.15–5.7% according to the comparison of numerical and experimental results, which proves the validity of the experimental results and reasonability of the calculated ice mechanics properties;(4)By comparing the waveform diagrams of the non-porous model and different porosity models, it can be found that the porosity affects the wave propagation velocity. By which, the greater the porosity, the slower the wave propagation velocity;(5)This research proves that the random pore model can simulate ice well. Additionally, by modifying the properties of the input model, more different ice models at different temperatures can be obtained, which provides a reliable modelling method for later research on ice. 


## Figures and Tables

**Figure 1 sensors-23-06045-f001:**
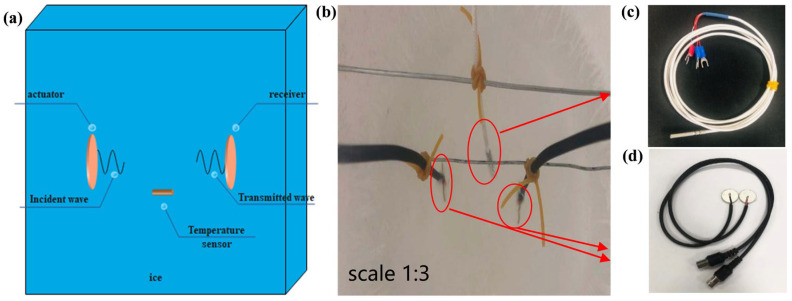
(**a**) Schematic diagram of ice sample structure; (**b**) ice specimen with sensors (scale 1:3); (**c**) the PZT; and (**d**) the temperature sensor.

**Figure 2 sensors-23-06045-f002:**
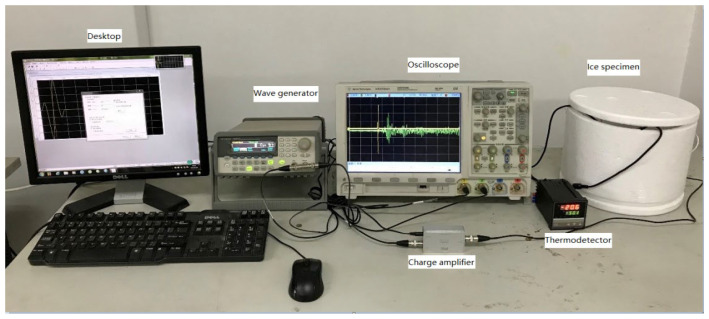
The experimental setup.

**Figure 3 sensors-23-06045-f003:**
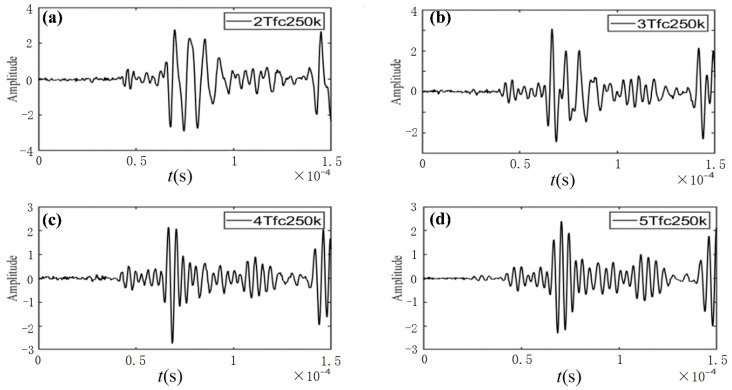
Measured ultrasonic signals with the same centre frequency (250 kHz) and various number of cycles: (**a**) *n* = 2, (**b**) *n* = 3, (**c**) *n* = 4, and (**d**) *n* = 5.

**Figure 4 sensors-23-06045-f004:**
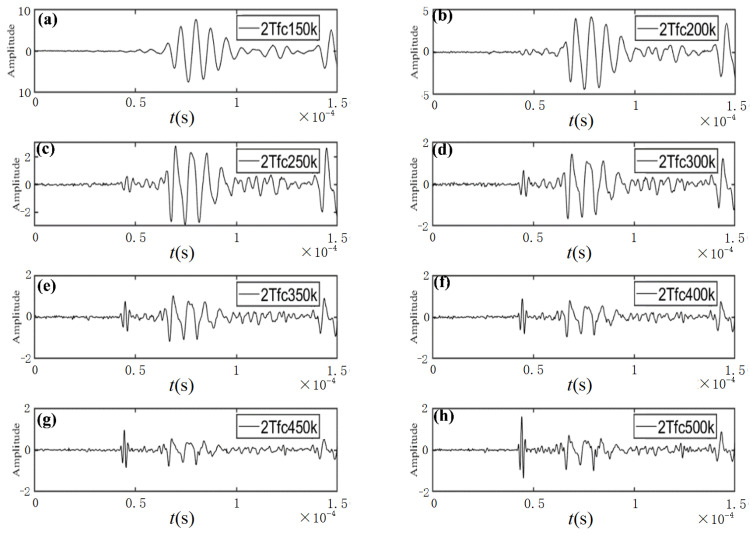
Measured ultrasonic signals with the same cycles (2T) and various central frequencies: (**a**) *f_c_* = 150 kHz, (**b**) *f_c_* = 200 kHz, (**c**) *f_c_* = 250 kHz, (**d**) *f_c_* = 300 kHz, (**e**) *f_c_* = 350 kHz, (**f**) *f_c_* = 400 kHz, (**g**) *f_c_* = 450 kHz, and (**h**) *f_c_* = 500 kHz.

**Figure 5 sensors-23-06045-f005:**
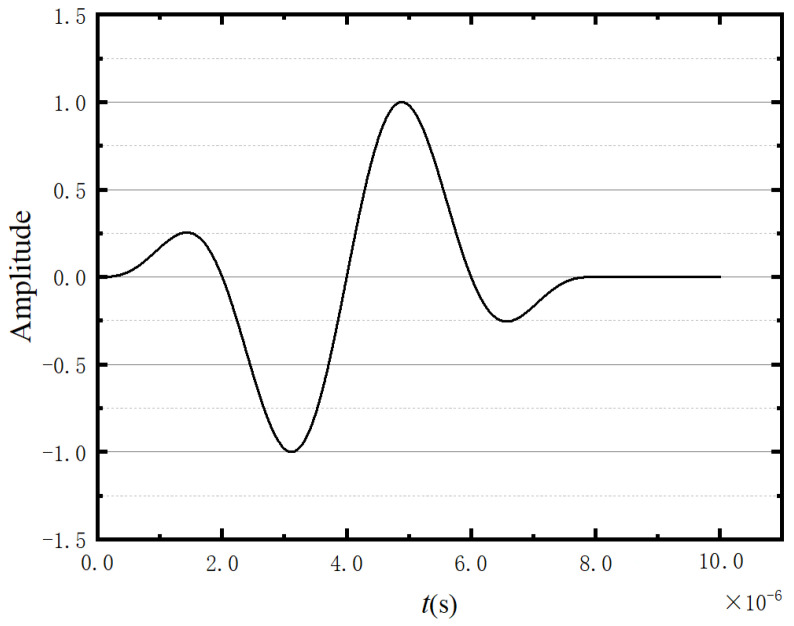
Waveform of the normalised 2-cycle 250 kHz tone burst signal.

**Figure 6 sensors-23-06045-f006:**
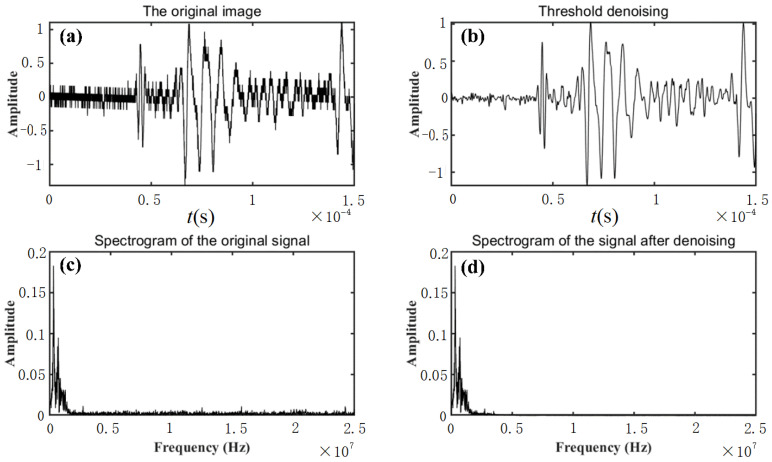
Time domain and spectrogram of the response signal: (**a**) original response signal, (**b**) signal after denoising, (**c**) spectrogram of the original signal, and (**d**) spectrogram of the signal after denoising.

**Figure 7 sensors-23-06045-f007:**
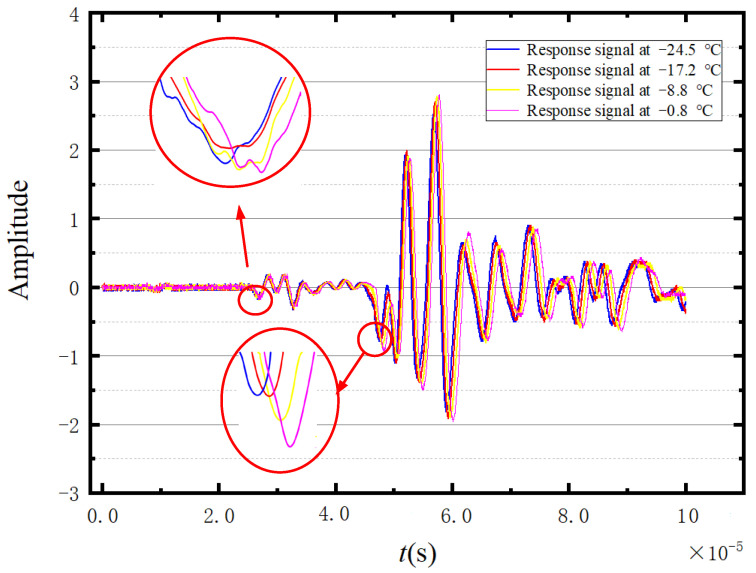
Four sets of measured waveforms.

**Figure 8 sensors-23-06045-f008:**
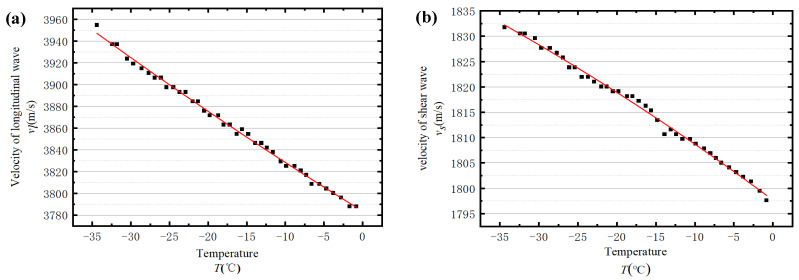
The relationship between wave velocity and temperature: (**a**) the relationship between longitudinal wave velocity and temperature and (**b**) the relationship between shear wave velocity and temperature.

**Figure 9 sensors-23-06045-f009:**
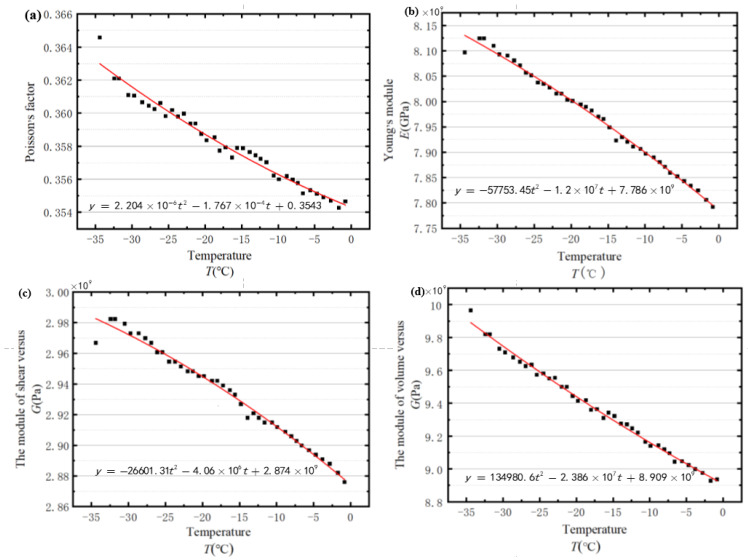
The changes of various properties with ice samples’ temperatures: (**a**) the Poisson’s factor of ice versus temperature of ice; (**b**) the Young’s module versus temperature of ice; (**c**) the module of volume versus temperature of ice; and (**d**) the module of shear versus temperature of ice.

**Figure 10 sensors-23-06045-f010:**
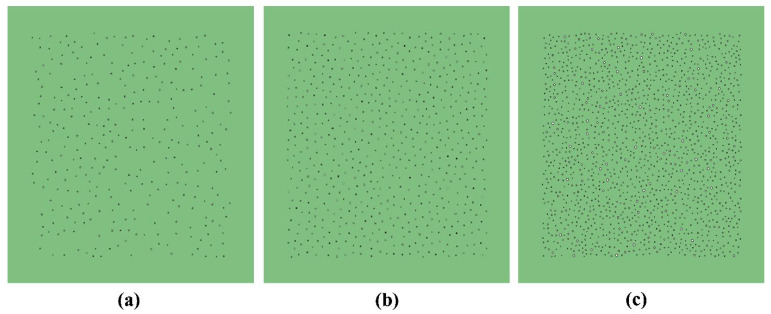
Numerical model of the ice specimen with different porosities: (**a**) 0.5%; (**b**) 1%; and (**c**) 3%.

**Figure 11 sensors-23-06045-f011:**
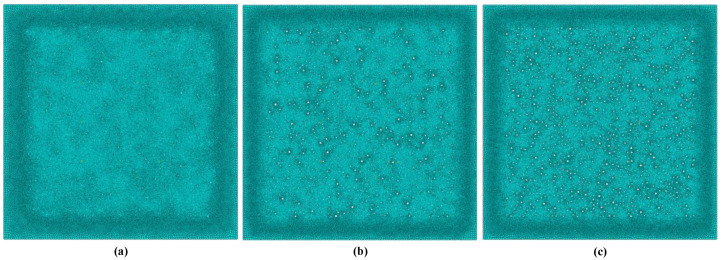
FEM meshes of the ice specimen with different porosities: (**a**) 0.5%; (**b**) 1%; and (**c**) 3%.

**Figure 12 sensors-23-06045-f012:**
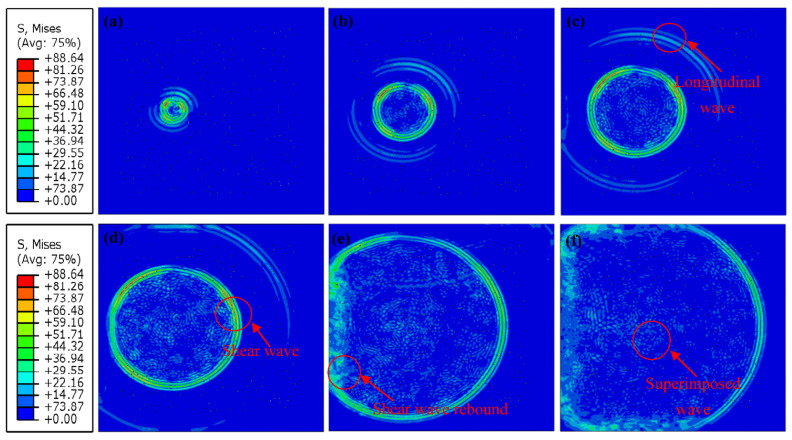
Wave propagation in a 2D ice model: (**a**–**f**) are the time slices of the wave propagation process: (**a**) time = 1.0 × 10^−5^ s, (**b**) time = 2.0 × 10^−5^ s, (**c**) time = 3.0 × 10^−5^ s, (**d**) time = 4.0 × 10^−5^ s, (**e**) time = 6 × 10^−5^ s, and (**f**) time = 7 × 10^−5^ s.

**Figure 13 sensors-23-06045-f013:**
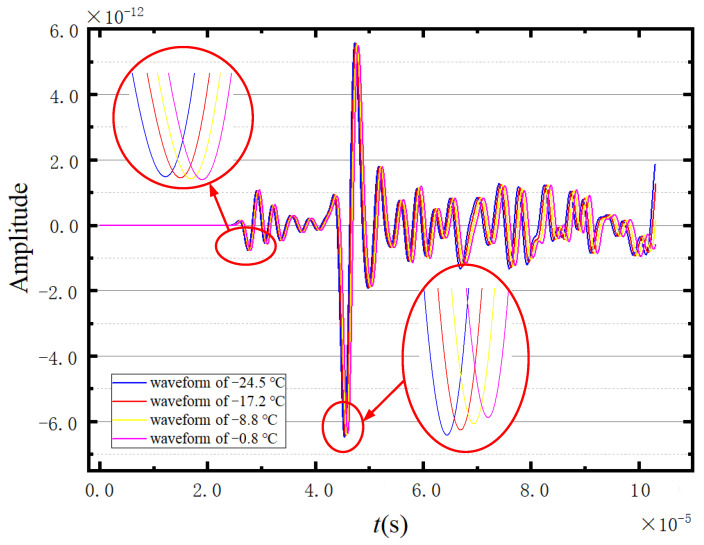
Four sets of numerical simulation waveforms.

**Figure 14 sensors-23-06045-f014:**
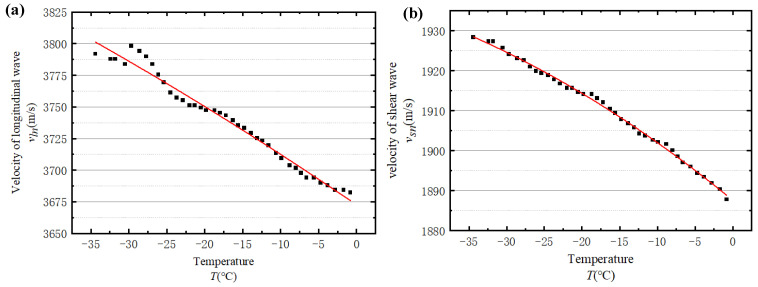
The relationship between wave velocities and temperatures: (**a**) the relationship between longitudinal wave velocity and temperature and (**b**) the relationship between shear wave velocity and temperature.

**Figure 15 sensors-23-06045-f015:**
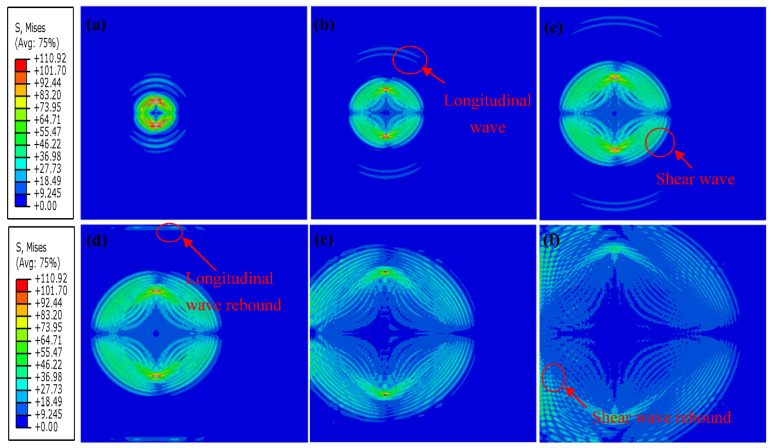
Wave propagation in a non-porous ice model and a 3% porosity ice model. (**a**–**f**) are the time slices of the wave propagation process: non-porous ice model (**a**) time = 2.0 × 10^−5^ s, (**b**) time = 3.0 × 10^−5^ s, (**c**) time = 4.0 × 10^−5^ s, (**d**) time = 4.5 × 10^−5^ s, (**e**) time = 6 × 10^−5^ s, and (**f**) time = 8 × 10^−5^ s.

**Figure 16 sensors-23-06045-f016:**
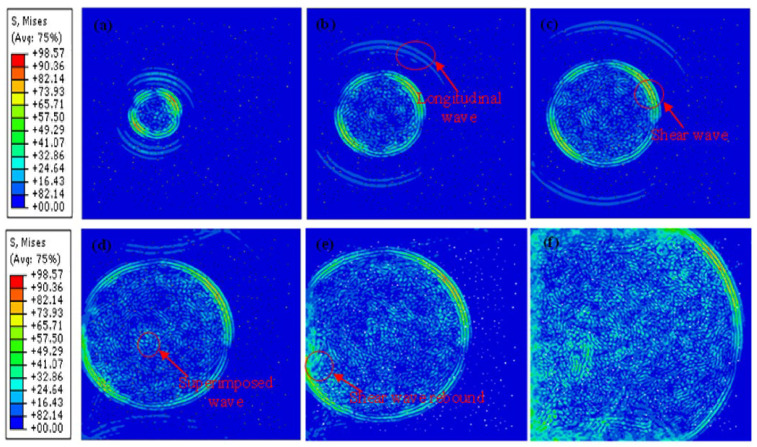
Wave propagation in a 3% porosity ice model, (**a**–**f**) are the time slices of the wave propagation process: (**a**) time = 2.2 × 10^−5^ s, (**b**) time = 3.2 × 10^−5^ s, (**c**) time = 4.0 × 10^−5^ s, (**d**) time = 5.0 × 10^−5^ s, (**e**) time = 5.8 × 10^−5^ s, and (**f**) time = 8 × 10^−5^ s.

**Figure 17 sensors-23-06045-f017:**
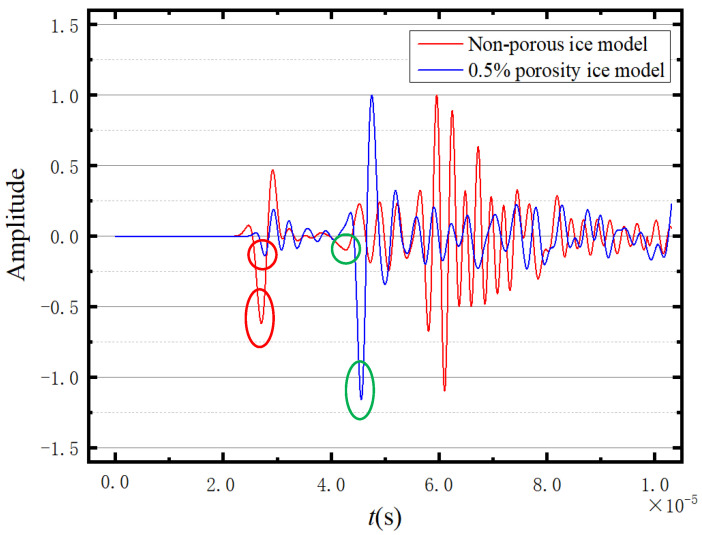
Comparison of the waveforms of the two models.

**Figure 18 sensors-23-06045-f018:**
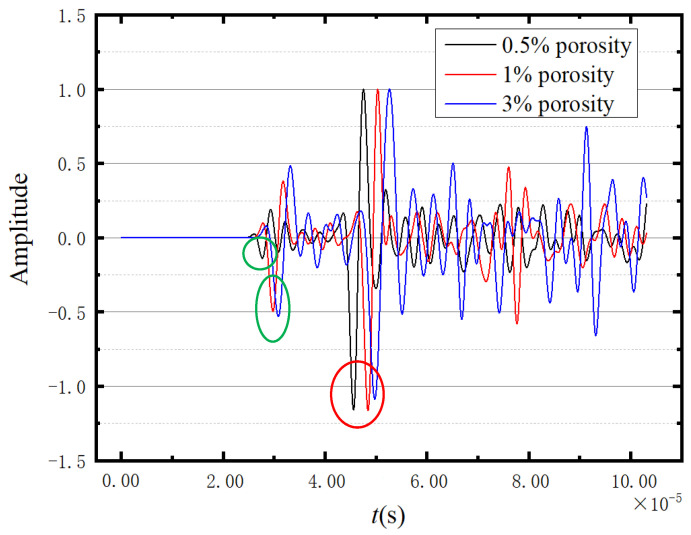
Comparison of three porosity waveforms.

**Figure 19 sensors-23-06045-f019:**
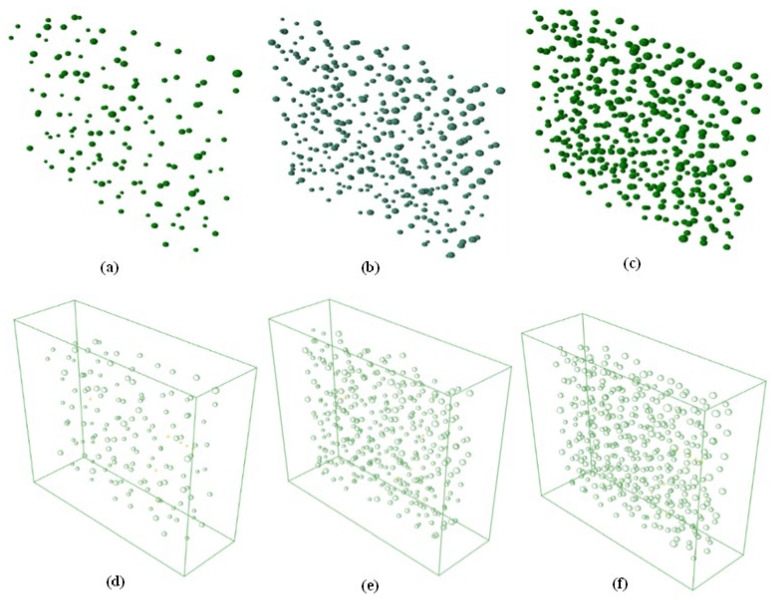
Three-dimensional numerical model: (**a**–**c**) the spherical pore of the three different porosities: (**a**) 0.5% porosity, (**b**) 1% porosity, and (**c**) 3% porosity; (**d**–**f**) three-dimensional ice random pore model of the three different porosities: (**d**) 0.5% porosity, (**e**) 1% porosity, and (**f**) 3% porosity.

**Figure 20 sensors-23-06045-f020:**
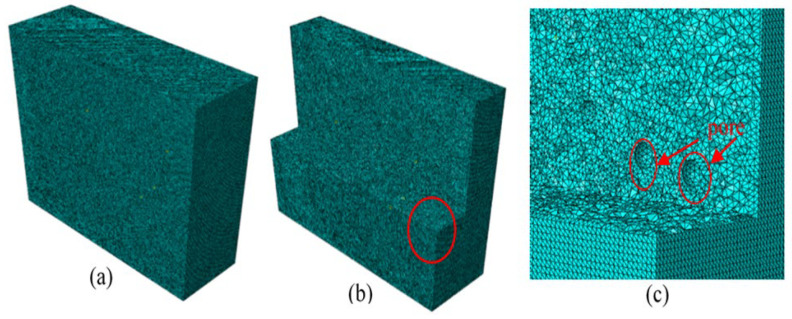
FEM meshes of the 3D ice model: (**a**) global view; (**b**) internal view; and (**c**) zoomed view of the red circle in (**b**).

**Figure 21 sensors-23-06045-f021:**
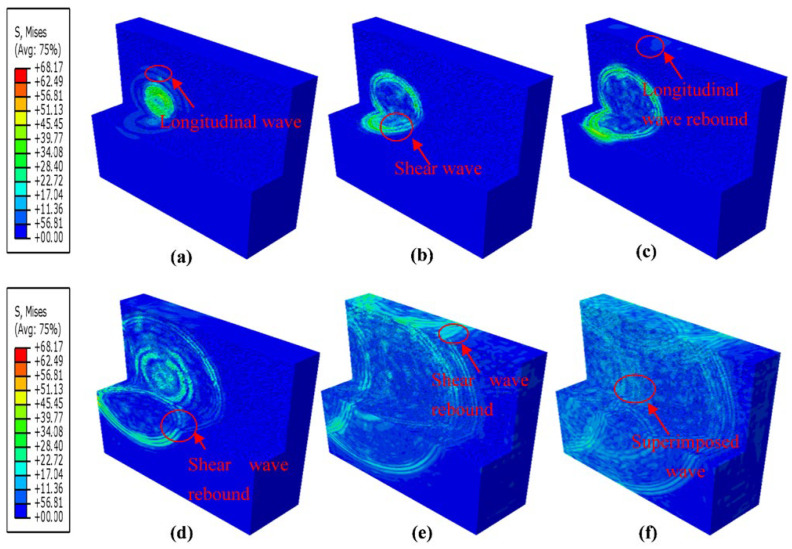
Wave propagation in the 3D ice model: (**a**–**f**) are the time slices of the wave propagation process: (**a**) time = 1.0 × 10^−5^ s, (**b**) time = 1.7 × 10^−5^ s, (**c**) time = 2.0 × 10^−5^ s, (**d**) time = 3.1 × 10^−5^ s, (**e**) time = 5.2 × 10^−5^ s, and (**f**) time = 6.3 × 10^−5^ s.

**Figure 22 sensors-23-06045-f022:**
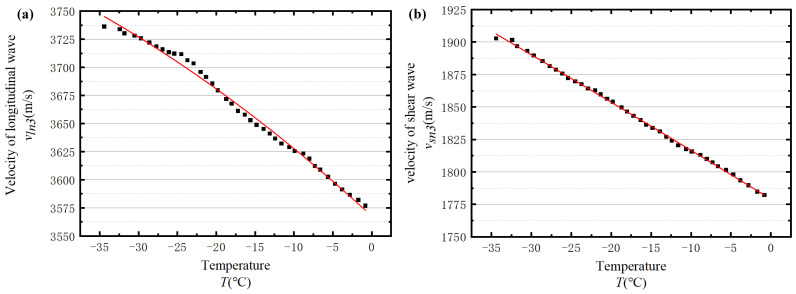
Three-dimensional numerical simulation of the relationship between wave velocities and temperatures: (**a**) the relationship between longitudinal wave velocity and temperature and (**b**) the relationship between shear wave velocity and temperature.

**Figure 23 sensors-23-06045-f023:**
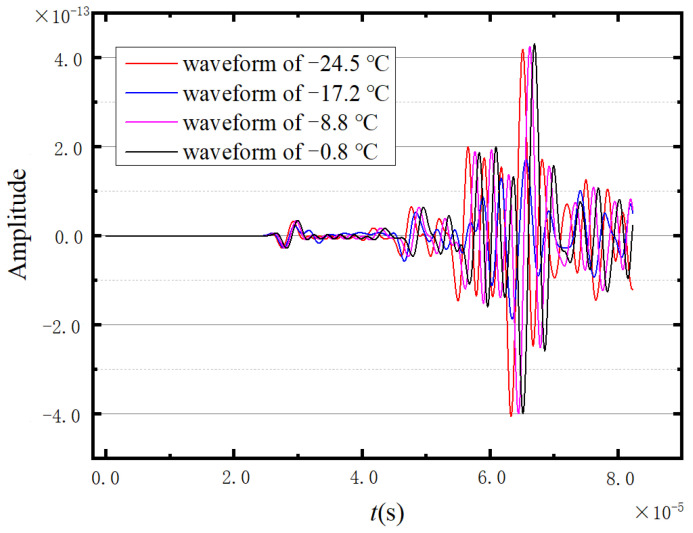
Combination chart of four sets of 3D numerical simulation.

**Figure 24 sensors-23-06045-f024:**
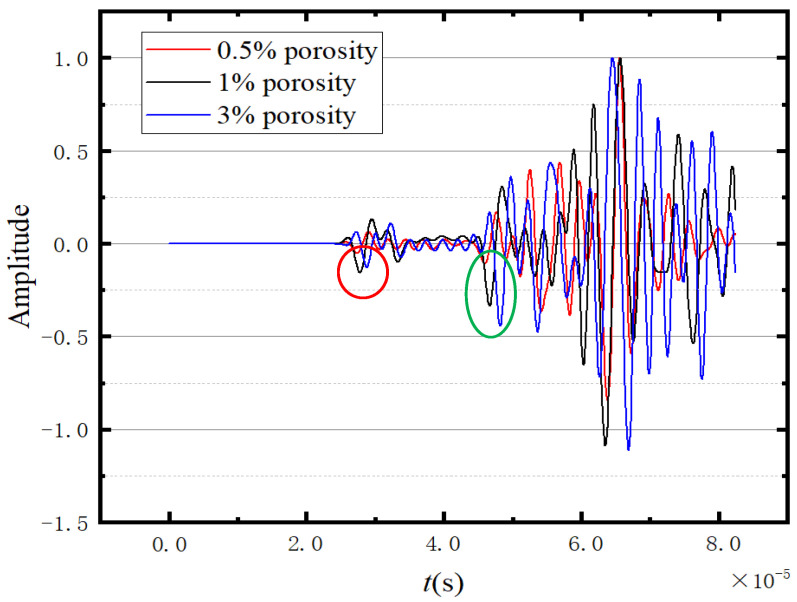
Comparison of three porosity waveforms of 3D models.

**Figure 25 sensors-23-06045-f025:**
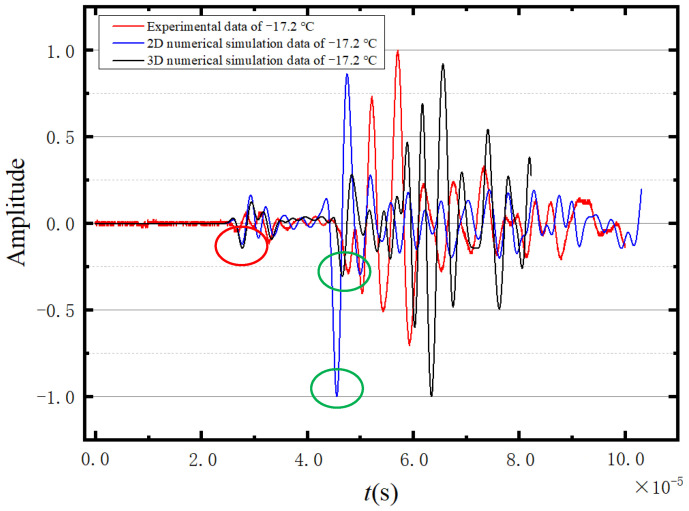
Comparison of experimental data and simulation.

**Table 1 sensors-23-06045-t001:** Dimensions and material properties of PZT.

Diameter	Thickness	d_33_	K^T3^
20 mm	1 mm	460 × 10^−12^ m/V	1700

**Table 2 sensors-23-06045-t002:** Values of the longitudinal sound velocity in ice by different authors.

Studies	Kupperman et al. [42]	Gammon et al. [43]	Randhawa [44]	Present
Sound velocity (m/s)	3950	3892–4040	3980	3790–3960

**Table 3 sensors-23-06045-t003:** Material mechanical properties of ice.

Temperature *T* (°C)	Density *ρ* (kg·m^3^)	Poisson’s Factor *μ*	Young’s Module *E* (GPa)	Shear Module *G* (GPa)	Volume Module *K* (GPa)
−24.5	890	0.360	8.0375	2.954	9.581
−17.2	890	0.358	7.982	2.939	9.363
−8.8	890	0.356	7.889	2.908	9.143
−0.8	890	0.355	7.792	2.876	8.935

**Table 4 sensors-23-06045-t004:** Wave velocity of tests and simulations and the errors between them.

Temperature *T* (°C)	Longitudinal Wave Velocity of the Test *V_l_* (m/s)	Two-Dimensional Simulated Longitudinal Wave Velocity *V_ln_*_2_ (m/s)	Errors of 2D Simulation Results (%)	Three-Dimensional Simulated Longitudinal Wave Velocity *V_ln_*_3_ (m/s)	Deviation of 3D Simulation Results (%)	Two-Dimensional Shear Wave Velocity of the Test *Vs* (m/s)	Two-Dimensional Simulated Shear Wave Velocity *V_sn_*_3_ (m/s)	Deviation of 2D Simulation Results (%)	Three-Dimensional Simulated Shear Wave Velocity *V_sn_*_3_ (m/s)	Errors of 2D Simulation Results (%)
−34.4	3954.80	3791.98	4.12	3736.21	5.53	1825.77	1928.37	5.62	1902.83	4.22
−32.4	3937.01	3787.88	3.79	3733.69	5.16	1830.54	1927.31	5.29	1901.53	3.88
−31.8	3937.01	3787.88	3.79	3730.15	5.25	1830.54	1927.31	5.29	1896.71	3.61
−30.5	3923.77	3783.78	3.57	3728.05	4.98	1829.59	1925.72	5.25	1893.21	3.48
−29.7	3919.37	3798.16	3.09	3725.75	4.94	1827.68	1924.13	5.28	1889.56	3.39
−28.6	3914.99	3794.04	3.09	3721.98	4.93	1827.68	1923.08	5.22	1885.24	3.15
−27.7	3910.61	3789.93	3.09	3718.66	4.91	1826.72	1922.55	5.25	1881.36	2.99
−26.9	3906.25	3783.78	3.14	3715.92	4.87	1825.77	1920.97	5.21	1878.64	2.90
−26.1	3906.25	3775.62	3.34	3713.45	4.94	1823.87	1919.91	5.27	1875.64	2.84
−25.4	3897.55	3769.52	3.28	3712.08	4.76	1823.87	1919.39	5.24	1872.22	2.65
−24.5	3897.55	3761.40	3.50	3711.56	4.77	1821.97	1918.90	5.30	1869.66	2.62
−23.7	3893.21	3757.38	3.49	3706.31	4.80	1821.97	1917.81	5.26	1867.54	2.50
−22.9	3893.21	3755.36	3.54	3703.33	4.88	1821.02	1916.76	5.26	1864.26	2.37
−22	3884.57	3751.34	3.43	3695.85	4.86	1820.07	1915.71	5.25	1862.74	2.34
−21.3	3884.57	3751.34	3.43	3691.33	4.97	1820.07	1915.71	5.25	1859.69	2.18
−20.5	3875.97	3749.33	3.27	3685.46	4.92	1819.13	1914.66	5.25	1856.24	2.04
−19.8	3871.68	3747.32	3.21	3679.51	4.96	1819.13	1914.14	5.22	1853.96	1.91
−18.7	3871.68	3747.32	3.21	3671.69	5.17	1818.18	1914.14	5.28	1849.54	1.72
−18	3863.13	3745.32	3.05	3667.52	5.06	1818.18	1913.09	5.22	1846.33	1.55
−17.2	3863.13	3743.32	4.00	3661.09	5.23	1817.24	1912.05	4.94	1843.08	1.42
−16.3	3854.63	3739.32	2.99	3657.74	5.11	1816.29	1910.48	5.19	1839.85	1.30
−15.6	3858.88	3735.33	3.20	3652.86	5.34	1815.35	1909.44	5.18	1836.34	1.16
−14.8	3854.63	3733.33	3.15	3648.69	5.34	1813.47	1907.88	5.21	1833.85	1.12
−13.9	3846.15	3729.36	3.04	3645.21	5.22	1810.66	1906.84	5.31	1831.02	1.12
−13.1	3846.15	3725.39	3.14	3640.98	5.33	1811.59	1905.80	5.20	1826.98	0.85
−12.4	3841.93	3723.40	3.09	3636.54	5.35	1810.66	1904.24	5.17	1823.87	0.73
−11.6	3837.72	3719.45	3.08	3632.23	5.35	1809.72	1903.73	5.19	1820.33	0.59
−10.6	3829.32	3713.53	3.02	3628.96	5.23	1809.72	1902.69	5.14	1817.52	0.43
−9.9	3825.14	3709.59	3.02	3625.43	5.22	1808.79	1902.17	5.16	1815.65	0.38
−8.8	3825.14	3703.70	3.17	3623.19	5.28	1807.85	1901.66	5.10	1813	0.28
−8	3820.96	3701.75	3.12	3618.65	5.29	1806.92	1900.11	5.16	1809.89	0.16
−7.3	3816.79	3697.83	3.12	3612.12	5.36	1805.99	1898.56	5.13	1807.35	0.08
−6.6	3808.49	3693.93	3.01	3608.94	5.24	1805.05	1897.02	5.10	1804.22	0.05
−5.6	3808.49	3693.93	3.01	3602.54	5.41	1804.12	1895.99	5.09	1801.36	0.15
−4.7	3804.35	3690.04	3.00	3596.43	5.47	1803.19	1894.45	5.06	1797.95	0.29
−3.8	3800.22	3688.09	2.95	3591.23	5.50	1802.27	1893.43	5.06	1793.43	0.49
−2.8	3796.10	3684.21	2.95	3586.54	5.52	1801.34	1891.89	5.03	1789.53	0.66
−1.7	3787.88	3684.21	2.74	3581.93	5.44	1799.49	1890.36	5.05	1784.63	0.83
−0.8	3787.88	3682.27	2.70	3576.9	5.57	1797.64	1887.81	5.02	1782.08	0.87

## Data Availability

The datasets used and/or analysed during the current study are available from the corresponding author on reasonable request.

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
