# Peer review of "Embedded Ultrasonic Inspection on the Mechanical Properties of Cold Region Ice under Varying Temperatures"

_sensors, 2023, doi:10.3390/s23136045_

Round 1

Reviewer 1 Report

Very interesting work. Minor revisions are in order.

Language needs to be revised significantly.

“ve test methods are destructive result in difficulties in obtaining the mechanical properties continuously under temperature-varying conditions for the sa”: why?

“the non-destructive testing technique makes it possible for real-time monitoring without impairing the performance of materials or components”: true for multiple techniques. Revise.

Fig 1: scales needed.

“a foam box to prohibit the fast temperature variation.”: was this confirmed?

“Naturally, ice contains entrapped air bubbles”: why? Always?

More details on the numerical simulations are needed.

revise

Author Response

Reply to reviewers

The authors would like to thank the reviewers for their accurate and insightful review of the manuscript. For easy perusal and re-review, the modified portions of the revised manuscript are highlighted in blue.

Reviewer #1

Very interesting work. Minor revisions are in order.

A: Thanks for the positive comments and suggestions. The manuscript has been revised according to the reviewer’s comments, with point-to-point responses as listed below.

Q1: Language needs to be revised significantly.

A1: Thanks for the positive suggestions. We have revised the language according to the reviewer's requirements.

Q2: “ve test methods are destructive result in difficulties in obtaining the mechanical properties continuously under temperature-varying conditions for the sa”: why?

A2: Thank you for pointing out this lack of clarity and relative clarifications are added as follows (on Page 1 line 44):

However, due to the brittleness of the ice material itself, fracture is inevitable in the experiment. Moreover, the time from the testing start to the fracture ending of is very short. Therefore, in a single experiment, it is not possible to obtain changes in ice properties caused by continuous temperature changes.

Q3: “the non-destructive testing technique makes it possible for real-time monitoring without impairing the performance of materials or components”: true for multiple techniques. Revise.

A3: Thanks for the reviewer’s valuable comments. We have revised this part as follows (on Page 2 line 49):

In order to obtain the mechanical properties of ice that continuously change with temperature without damaging the existing ice structure, a non-destructive testing method is needed, which can achieve real-time monitoring and is suitable for ice materials.

Q4:Fig 1: scales needed.

A4: Thanks for the reviewer’s valuable comments. Following your suggestion, we have added a scale of Figure 1 in the resubmitted manuscript.

Q5: “a foam box to prohibit the fast temperature variation.”: was this confirmed?

A5: Thank you for pointing out the omission. The foam box, as a common thermal insulation material in daily life, is used to slow down the impact of external temperature on water, thus slowing down the speed of water icing. If the freezing speed is too fast, it can cause the ice surface to expand and crack. Cracking ice will affect the experimental results.

Q6: “Naturally, ice contains entrapped air bubbles”: why? Always?

A6: Thank you for pointing out this lack of clarity. In nature, ice contains pores, which are related to the growth process of ice. The formation of ice from water is a process from the outside to the inside. After the surface freezes, the air in the water cannot pass through the ice, forming pores inside the ice. These pores vary in size and are randomly located, so randomly distributed circular pores were used in the manuscript to simulate the pores inside the ice.

Q7:More details on the numerical simulations are needed.

A7: Thank you for pointing out this lack of clarity and relative clarifications are added as follows (on Page 11 line 331):

In the numerical model, the ice is treated as a linear small without considering the nonlinear behavior and tensile or compressive damage. The dimensions of tested specimens are 210×210 mm2 and the pores (air bubbles) are concentrated within a central region of 180×180 mm2. Additionally, the round pores in 2D model are utilized. They are classified into small pores having diameters of 0.5 ~ 1 mm and large pores having diameters of 1 - 1.5 mm, with the volume ratios being 6:4. Moreover, the pores in the ice model are reciprocally independent with the distances between the pores greater than 1.5 mm. In the modeling process, random pores are first generated using MATLAB, and then the generated pore model is imported into the ice model built in ABAQUS. The final model is obtained by cutting out random pores from the original ice model. In the model, the excitation of the PZT actuator was simplified by applying a concentrated force at the same position as the PZT sensor. The concentrated force is generated by the same 2-cycle 250 kHz tone burst signal as the experimental test (as shown in Figure 5). In addition, the distance between the actuator and receiver is 70mm. In the model, the boundaries around the ice are the boundaries of wave propagation.

Reviewer 2 Report

Comment 1. The title of the article, abstract and content are consistent.

Comment 2. Line 111, What is the measurement range of the sensor -50 to -200 or -50 to +200? I would like to ask you to indicate the measurement range clearly.

Comment 3. Comprehensible starting points, description of the experiment, clear evaluation.

Comment 4. To equations (1) – (6), add explanations for the symbols: G, ρ, K. In equation (7), the quantity t.

Comment 5. Provide an explanation of the texts "2Tfc150k, ..... and so on" in images 3 and 4.

Comment 6. State the scheme or sensor distance value.

Comment 7. Line 290, mm2, Line 373 mm3 and elsewhere, is it the correct size?

Comment 8. Justify according to which criteria they are classified: small pores in the diameter range of 0.5-1 mm and large pores as 1-1.5 mm?

Comment 9. Justify the ratio 6:4 for the volume ratios.

Comment 10. Figure 13. In the text for Figure 13 you write that only the speed of the wave is affected by the changing temperature and the change in the shape of the wave can be ignored. In figure 13, the speed does not find (but the time t?), get it in right. State on ewhat basis of which the change in waveform can be ignored?

Comment 11. Line 345 Explain the choice of temperature -17.2℃ for numerical simulations.

Comment 12. State the used simulation software.

Comment 13. Your results show relative errors in the range of 0.15% - 5.7%. Can a relative error of 5.7% be considered acceptable, and can you explain the origin of this error?

Author Response

The authors thank the reviewer for the careful reading of our manuscript and all the concerns are addressed point by point in the following

Q1. The title of the article, abstract and content are consistent.

A1: Thank you for confirming of our manuscript.

Q 2. Line 111, What is the measurement range of the sensor -50 to -200 or -50 to +200? I would like to ask you to indicate the measurement range clearly.

A2: Thank you for pointing out this lack of clarity and relative clarifications are added as follows (on Page 3 Lines 121):

Its measurement range is -50 ℃ ~ +200 ℃.

Q 3. Comprehensible starting points, description of the experiment, clear evaluation.

A3: Thanks for the positive comments.

Q 4. To equations (1) – (6), add explanations for the symbols: G, ρ, K. In equation (7), the quantity t.

A4: Thank you for pointing out this lack of clarity. Following the reviewer’s suggestion, relative clarifications are added as follows (on Page 4 line 172 and Page 5 line 187):

In the equation (1) ~ (6), ρ is the density, E is Young’s modulus,  is Shear modulus,  is Volume modulus. In equation (7),  is time.

Q 5. Provide an explanation of the texts "2Tfc150k, ..... and so on" in images 3 and 4.

A5: This is a very good point raised by the reviewer. In which, 2T means the ultrasound signal is a two cycle signal, and fc is the center frequency of the signal. Following the reviewer’s suggestion, we improved this part as follows (on Page 6 line 227and Page 7 line 238):

Figure 3. Measured ultrasonic signals with same center frequency (250kHz) and various number of cycles: (a) n=2, (b) n=3, (c) n=4, (d) n=5.

Figure 4. Measured ultrasonic signals with same cycles (2T) and various central frequencies: (a) fc=150 kHz, (b) fc=200 kHz, (c) fc=250 kHz, (d) fc=300 kHz, (e) fc=350 kHz, (f) fc=400 kHz, (g) fc=450 kHz, (h) fc=500 kHz.

Q 6. State the scheme or sensor distance value.

A6:Thank you for pointing out this point. Following the reviewer’s suggestion, relative clarifications are added as follows (on Page 4 line 144):

PZTs and temperature sensor are set in the middle of the ice specimen. The distance between the two PZTs is considered to be 70 mm. And the temperature sensor is set at the uniform level of the PZTs due to the effect of temperature stratification in ice.

Q 7. Line 290, mm2, Line 373 mm3 and elsewhere, is it the correct size?

A7:Thank you for the comment. Iine 290 is a 2D model, so the unit is mm2, and line 373 is a 3D model, so the unit is mm3.

Q 8. Justify according to which criteria they are classified: small pores in the diameter range of 0.5-1 mm and large pores as 1-1.5 mm?

A8: Thank you for pointing out this part. Before establishing the model, we observed and measured a large number of pores in natural and artificial ice. These pores, except for those larger than 5mm on the surface of the ice, are very small in size inside the ice, some of which cannot be measured using conventional measurement methods. In order to restore the uniformly distributed pores accumulated in the ice, we fixed the porosity during simulation, selected the distribution range of most pore sizes, and calculated the appropriate pore distribution using MATLAB.

Q 9. Justify the ratio 6:4 for the volume ratios.

A9: Thank you for the comment. This volume ratio was obtained through extensive testing, which not only meets the requirements of achieving porosity within the model, but also achieves uniform distribution of pores within the model.

Q10. Figure 13. In the text for Figure 13 you write that only the speed of the wave is affected by the changing temperature and the change in the shape of the wave can be ignored. In figure 13, the speed does not find (but the time t?), get it in right. State on ewhat basis of which the change in waveform can be ignored?

A10: Thanks for this suggestions. As you can see in Figure 13, there is no significant change in the morphology of the waves at the four temperatures. If the image is not enlarged, it is difficult to detect the difference between the four curves. So we believe that the change in temperature has little effect on the waveform. However, in the red dashed circle in Figure 13, the results of amplifying the peaks show that the time of the peaks appearing in the four curves is different, which means that the wave propagation speeds corresponding to the four temperatures are different (because the distance between sensors remains constant). Correspondingly, we calculated the wave velocity at other temperatures.

Q 11. Line 345 Explain the choice of temperature -17.2℃ for numerical simulations.

A11: Thank you for the valuable comment. The result of -17.2 ℃ was randomly selected to demonstrate the process of wave propagation in ice in numerical simulations. Of course, other temperature results can also be selected for display. The selection of different temperatures has no effect on the propagation process of waves in ice.

Q 12. State the used simulation software.

A12: Thank you for pointing out this lack of clarity and relative clarifications are added as follows (on Page 11 line 347):

In the modeling process, random pores are first generated using MATLAB, and then the generated pore model is imported into the ice model built in ABAQUS. The final model is obtained by cutting out random pores from the original ice model.

Q 13. Your results show relative errors in the range of 0.15% - 5.7%. Can a relative error of 5.7% be considered acceptable, and can you explain the origin of this error?

A13: This point raised by the reviewer is very important. As we can see, the propagation speed of waves in ice is very fast. The distance between the two sensors is only 70mm, so the propagation time is very short. However, even a small-time difference can lead to significant differences in wave velocity. Therefore, we believe that 5.7% is not a significant error and is acceptable. In addition, we believe that the main reason for the difference between the experimental and simulation results is due to material parameters and boundary conditions.

Reviewer 3 Report

This study proposed a sophisticated embedded ultrasonic system to inspect the mechanical properties of ice in real time. This embedded ultrasonic system provides a platform to acquire the response of the ice continuously. With this system, the ice properties under specific temperature conditions are identified based on the intrinsic relationships between the wave propagation velocities and the mechanical properties of ice. Also, the feasibility and effectiveness of the proposed system-based method are validated via the ultrasonic experiments and numerical simulations.

1 This study proposed a ultrasonic system to monitore the the mechanical properties of ice. The ultrasonic monitoring technology has been developed well in recent years, so what is the main innovation point of this research. Please emphasize the innovation in the abstract

2 In Figure 6, there is little difference between spectrogram before and after noise reduction, so why the noise reduction operation is carried out?

3 There exist many typos, e.g., Figure9 to Figure 9. Please proofread the whole manuscript for the modification.

4 The quality of the pictures in the paper needs to be further improved, specifically, including the clarity of the pictures and the annotations in the pictures should be further improved. For instance, the horizontal and vertical coordinates in Figures 3 and 4 are too vague.

5 For many figures, the scaling in both directions, e.g., length and height, is inconsistent.

 6 In this study, after obtaining the monitoring data, the corresponding data modeling and data analysis were carried out, including regression analysis and spectrum analysis. However, data modeling and analysis related reviews are not mentioned in the introduction review, so it is suggested that the author add some literatures on data modeling and analysis, e.g.,  "Monitoring, Analyses and Modelling" "Towards high-precision data modeling of SHM measurements using an improved sparse Bayesian learning scheme with strong generalization ability""Modelling and forecasting of SHM strain measurement for a large-scale suspension bridge during typhoon events using variational heteroscedasic Gaussian process" "Bayesian dynamic linear model framework for SHM data forecasting and missing data imputation during typhoon events". Througth these literatures, the readers can better understand the data analysis part of this study. 

7 In principle, 3D models can better simulate reality, and the results should be more accurate. The authors made 2D and 3D model simulation analysis. Generally, 3D model analysis is enough to get accurate analysis results, why do 2D simulation analysis is needed?

8 Figure 17, generally the amplitude of non-porous ice model  is larger than that of porosity ice. But during 0.4s and 0.5s, the amplitude of non-porous ice model  is smaller, compared with the prosity ice. Please clarify this phenomenon.

Moderate editing of English language required

Author Response

This study proposed a sophisticated embedded ultrasonic system to inspect the mechanical properties of ice in real time. This embedded ultrasonic system provides a platform to acquire the response of the ice continuously. With this system, the ice properties under specific temperature conditions are identified based on the intrinsic relationships between the wave propagation velocities and the mechanical properties of ice. Also, the feasibility and effectiveness of the proposed system-based method are validated via the ultrasonic experiments and numerical simulations.

A: Thanks for the positive comments and suggestions. The manuscript has been revised according to the reviewer’s comments, with point-to-point responses as listed below.

Q1: This study proposed a ultrasonic system to monitor the mechanical properties of ice. The ultrasonic monitoring technology has been developed well in recent years, so what is the main innovation point of this research. Please emphasize the innovation in the abstract

A1: Thank you for pointing out this lack of clarity. This study proposes an embedded ultrasound system for online real-time detection of ice mechanical properties. Compared to ordinary ultrasonic testing equipment, this system uses processed pzt as the excitation and reception sensor, which reduces the impact of traditional sensor volume on the performance testing results of the material itself and provides conditions for the ultrasonic system to be embedded in the ice. It has also been confirmed through experiments that this embedded ultrasonic system can obtain the mechanical parameters of ice that continuously change with temperature.

Q2: In Figure 6, there is little difference between spectrogram before and after noise reduction, so why the noise reduction operation is carried out?

A2: Thank you for pointing out this lack of clarity and relative clarifications are added as follows:

As you mentioned, there is little difference between spectrogram before and after noise reduction, but noise reduction makes the selected peak more accurately. We believe that even if the signal difference after noise reduction is not significant, it is still necessary. This is because in the experiment, the distance between the two sensors inside the ice is small, and the speed of wave propagation is fast, which requires high accuracy in the peak position of the wave. After all, a small positional difference can lead to significant differences in wave velocity calculation results.

Q3: There exist many typos, e.g., Figure9 to Figure 9. Please proofread the whole manuscript for the modification.

A3: Thank you for pointing out this error. I am very sorry for the occurrence of such an error in the text. Therefore, I have carefully proofread the whole manuscript.

Q4: The quality of the pictures in the paper needs to be further improved, specifically, including the clarity of the pictures and the annotations in the pictures should be further improved. For instance, the horizontal and vertical coordinates in Figures 3 and 4 are too vague.

A4: Thanks for the reviewer’s positive suggestions. We have made modifications to the images as requested by the reviewer, including other images in the manuscript. The modified image is presented in the resubmitted manuscript.

Q5: For many figures, the scaling in both directions, e.g., length and height, is inconsistent.

A5: Thanks for the reviewer’s valuable comments. We have made modifications according to the reviewer's requirements and improved the widths and heights scaling ratio of all image in the manuscript.

Q6: In this study, after obtaining the monitoring data, the corresponding data modeling and data analysis were carried out, including regression analysis and spectrum analysis. However, data modeling and analysis related reviews are not mentioned in the introduction review, so it is suggested that the author add some literatures on data modeling and analysis, e.g.,  "Monitoring, Analyses and Modelling" "Towards high-precision data modeling of SHM measurements using an improved sparse Bayesian learning scheme with strong generalization ability""Modelling and forecasting of SHM strain measurement for a large-scale suspension bridge during typhoon events using variational heteroscedasic Gaussian process" "Bayesian dynamic linear model framework for SHM data forecasting and missing data imputation during typhoon events". Througth these literatures, the readers can better understand the data analysis part of this study. 

A6: This is a very good point raised by the reviewer and the authors agree with the point. Following the reviewer’s suggestion, we add some literatures on data modeling and analysis in the introduction review(on Page 2 line 72).

Q7: In principle, 3D models can better simulate reality, and the results should be more accurate. The authors made 2D and 3D model simulation analysis. Generally, 3D model analysis is enough to get accurate analysis results, why do 2D simulation analysis is needed?

A7: Thanks for your comments. We agree with the reviewer's point of view. 3D models can indeed better simulate reality, but we still believe that 2D models are necessary. Firstly, by using a simple 2D model, the boundary rebound of waves during propagation is reduced compared to 3D models, which allows a clearer understanding of the propagation process of waves in ice; Secondly, 2D models can serve as a reference for 3D models, which provide a basis for the correctness of 3D results. Therefore, from the perspective of completeness and rigor in research, 2D models are still meaningful.

Q8: Figure 17, generally the amplitude of non-porous ice model is larger than that of porosity ice. But during 0.4s and 0.5s, the amplitude of non-porous ice model is smaller, compared with the porosity ice. Please clarify this phenomenon.

A8: This is a very good point raised by the reviewer and the authors agree with the point. Following the reviewer’s suggestion, we improved the relative discussion as follows (on Page 15 line 429):

In Figure 17, the amplitude in the first wave packet of the non-porous ice model is greater than that of the porous ice model. But, at the range of 0.4 ~ 0.5 s, the amplitude of the porous ice model is greater than that of the non-porous ice model. This is because the pore size in the porous model is 0.5 ~ 1 mm, which is much smaller than the wavelength (about 1500 m). Therefore, waves will form multiple reflections, refractions, and diffraction superposition under the rebound effect of the pore wall.

Round 2

Reviewer 1 Report

accept

na

Reviewer 3 Report

No further comments